# Gait Variability to Phenotype Common Orthopedic Gait Impairments Using Wearable Sensors

**DOI:** 10.3390/s22239301

**Published:** 2022-11-29

**Authors:** Junichi Kushioka, Ruopeng Sun, Wei Zhang, Amir Muaremi, Heike Leutheuser, Charles A. Odonkor, Matthew Smuck

**Affiliations:** 1Department of Orthopaedic Surgery, Stanford University, Stanford, CA 94305, USA; 2Division of Physical Medicine and Rehabilitation, Stanford University, Stanford, CA 94305, USA; 3Laboratory of Movement Analysis and Measurement, Ecole Polytechnique Fédérale de Lausanne (EPFL), 1015 Lausanne, Switzerland; 4Novartis Institutes for BioMedical Research, 4056 Basel, Switzerland; 5Machine Learning and Data Analytics Lab (MaD Lab), Department Artificial Intelligence in Biomedical Engineering (AIBE), Friedrich-Alexander-Universität Erlangen-Nürnberg (FAU), 91052 Erlangen, Germany; 6Department of Orthopedics and Rehabilitation, Division of Physiatry, Yale School of Medicine, New Haven, CT 06510, USA

**Keywords:** lumbar spinal stenosis, knee osteoarthritis, wearable IMU sensor, gait variability, gait impairment

## Abstract

Mobility impairments are a common symptom of age-related degenerative diseases. Gait features can discriminate those with mobility disorders from healthy individuals, yet phenotyping specific pathologies remains challenging. This study aims to identify if gait parameters derived from two foot-mounted inertial measurement units (IMU) during the 6 min walk test (6MWT) can phenotype mobility impairment from different pathologies (Lumbar spinal stenosis (LSS)—neurogenic diseases, and knee osteoarthritis (KOA)—structural joint disease). Bilateral foot-mounted IMU data during the 6MWT were collected from patients with LSS and KOA and matched healthy controls (N = 30, 10 for each group). Eleven gait parameters representing four domains (pace, rhythm, asymmetry, variability) were derived for each minute of the 6MWT. In the entire 6MWT, gait parameters in all four domains distinguished between controls and both disease groups; however, the disease groups demonstrated no statistical differences, with a trend toward higher stride length variability in the LSS group (*p* = 0.057). Additional minute-by-minute comparisons identified stride length variability as a statistically significant marker between disease groups during the middle portion of 6WMT (3rd min: *p* ≤ 0.05; 4th min: *p =* 0.06). These findings demonstrate that gait variability measures are a potential biomarker to phenotype mobility impairment from different pathologies. Increased gait variability indicates loss of gait rhythmicity, a common feature in neurologic impairment of locomotor control, thus reflecting the underlying mechanism for the gait impairment in LSS. Findings from this work also identify the middle portion of the 6MWT as a potential window to detect subtle gait differences between individuals with different origins of gait impairment.

## 1. Introduction

Mobility limitations are increasing with the aging of modern society [1,2]. Among those suffering from mobility impairment, lumbar spinal stenosis (LSS) and knee osteoarthritis (KOA) are two of the most common causes worldwide [3,4,5], creating a substantial and increasing burden on society and the economy [6,7,8]. LSS has a neurogenic disease mechanism characterized by claudication, described as pain and/or numbness in the lower back and extremities that worsens with walking distance [9]. Alternatively, KOA has a structural joint disease mechanism characterized by knee pain from the initiation of walking [10], often associated with a “gelling phenomenon” or a feeling of joint stiffness that gradually mitigates with continuous movement [11]. Since the pathophysiology and symptomatology of LSS differ from KOA, the manifestations of their respective gait disturbances may also differ. Our previous analysis of walking tests, including the 6 min walk test (6MWT), identified significant differences between these two disorders and the healthy controls [12]. However, a challenge remains to further phenotype and distinguish between these two common causes of mobility impairment using gait measures.

Numerous parameters are used to characterize gait disturbances [13,14,15]. These gait parameters can be classified into four domains: rhythm, pace, asymmetry, and variability [16,17]. The rhythm domain represents measures of absolute timing. The pace domain relates to walking speed and length measures in the sagittal plane. The asymmetry domain links to differences between the right and left lower limb parameters, and the variability domain reflects fluctuations in spatiotemporal characteristics between steps. These gait parameters are often reported as mean values during the entire period of the test, however, recent work showed that the temporal change in gait performance over a period of time may reveal additional subtle gait alterations that may be fatigue-induced and/or disease-specific [18,19].

Traditionally, gait analysis has been conducted in well-equipped labs using 3D motion capture techniques, requiring the use of costly equipment and specialized facilities. Newer technologies (including inertial measurement units (IMU), smartphones, low-cost video/depth cameras, pressure sensors, ambient sensors, etc.), provide alternatives that can efficiently capture and analyze movement data in various settings [20]. Among these technologies, wearable IMUs are most commonly used as the most portable and flexible option to measure mobility [21], allowing physicians, physical therapists, and researchers to analyze gait in research laboratories, clinics, and in a patient’s home or community. IMU sensors contain small 3D accelerometers (linear acceleration), gyroscopes (angular velocity), and/or magnetometers (magnetic field) that can be used collectively to quantify movement through various time- and frequency-domain parameters [20]. By securely attaching wearable IMU sensors to various body segments, different kinematic and biomechanical parameters can be obtained with similar validity and reliability to conventional lab-based systems [22,23]. Gait features derived from wearable sensors have demonstrated potential clinical utility in neurological diseases by distinguishing disease populations from healthy controls and stratifying disease severity [16,18]. However, such clinical utility has yet to be demonstrated in orthopedic research.

This study aims to analyze data from foot-mounted IMUs worn by patients with LSS and KOA during the 6MWT to investigate gait characteristics in each of the four gait domains (rhythm, pace, asymmetry, and variability). Specifically, our primary goal is to extract gait features by each minute of the walking test to compare LSS and KOA and determine whether time series data reveals phenotypic differences between the two disease groups.

## 2. Materials and Methods

### 2.1. Participants

A total of 30 participants were enrolled in the study, with 10 in each of the 3 groups: LSS, KOA, and matched healthy controls (HC). The full study inclusion/exclusion criteria are detailed in a previous paper summarizing the overall results from multiple walking tests [12]. The study was HIPAA compliant and approved by the ethical committee for Human Subjects Research at Stanford University. Each participant provided written informed consent and completed the 36-Item Short-Form health survey (SF-36) [24].

### 2.2. Gait Analysis

Each participant underwent a series of gait assessments, including the 40 m fast-paced walk test [25], the self-paced walk test [26], and the 6MWT [25]. In this study, we focus on the 6MWT for its ability to assess endurance and change in gait parameters over time [25,27,28]. A minute-by-minute analysis of the 6MWT provides a unique opportunity to evaluate dynamic changes that occur during walking and investigate time series differences between the two disease groups [18,19].

During the entire 6MWT, participants wore two Shimmer 3 IMU sensors (Shimmer Sensing, Dublin, Ireland) on the dorsal surface of both feet. We used previously validated algorithms to extract spatial–temporal gait parameters that reflect steady-state walking (after excluding all turning episodes) [21]. For details on gait parameter extraction, please see Appendix A. Eleven gait parameters were extracted representing the four different gait domains as described in previous research [16,17], including the rhythm domain (cadence, double support duration, and swing phase duration), the pace domain (stride length and speed), the asymmetry domain (asymmetry index for stride length and swing duration), and the variability domain (coefficient of variation for double support duration, swing duration, stride length, and gait cycle duration). Detailed descriptions of the extracted parameters are listed in Table 1. Among the extracted parameters, the asymmetry index was calculated using the difference ratio of the parameters derived from each limb [21]. A greater asymmetry index indicates a higher gait asymmetry [21]. In contrast, the coefficient of variation (CV) is a measure of variability, which refers to the ratio between the standard deviation and the mean of each parameter, expressed as a percentage [29]. A higher gait variability (more outstanding CV) indicates a worsening gait consistency [30]. To avoid redundancy, we did not present parameters that demonstrated similar results within the same domain. This approach for gait analysis was used previously in multiple studies [18,30,31,32,33,34]. The extracted gait parameters were further divided into 1 min segments to analyze minute-by-minute changes over the entire 6MWT.

All participants identified the side with more/most pain. However, since our preliminary analysis revealed no significant differences between limbs for all extract gait parameters, we reported averaged data for both limbs in this work. For data comparisons between limbs, please refer to Appendix A.

### 2.3. Statistical Methods

All derived gait parameters were checked for normal distribution using the Shapiro–Wilk test. Gait variables that were not normally distributed (variability and asymmetry domain measures) were log-transformed. Differences in parameters among the three groups from the entire 6MWT were analyzed by a one-way ANOVA test followed by Tukey’s test. Next, differences in parameters among three groups in each minute of the 6MWT were analyzed by two-way ANOVA with group and minute as factors. In addition, differences in parameters between the LSS and KOA groups were analyzed for each minute of the 6MWT using the Student T-test. A *p*-value < 0.05 was defined as statistically significant. All sensor data processing and gait feature extractions were performed using customized MATLAB (MathWorks, Inc., Natick, MA, USA) programs, and data visualization and statistical analyses were performed using customized Python programs.

## 3. Results

### 3.1. Participants’ Characteristics

Participants’ demographic characteristics are presented in Appendix A. The average age of all study participants was 65 years, with a range of 46–86 and an average BMI of 30.3. Additional details on the participants’ demographic and clinical characteristics were provided in the previous summary paper [12].

### 3.2. Analysis of the Entire 6MWT

Figure 1 and Table 2 summarize the gait parameters extracted from the analysis of the entire 6MWT. In the rhythm domain, LSS and KOA groups showed a significantly lower cadence and swing ratio and a significantly higher double support ratio than the HC group (HC-LSS: *p* < 0.05, HC-KOA: *p* < 0.05), indicating decreased gait rhythm. In the pace domain, LSS and KOA showed significantly lower speed than HC (HC-LSS: *p* < 0.05, HC-KOA: *p* < 0.05). While both disease groups showed lower stride length, the difference between LSS and HC did not reach statistical significance (HC-LSS: *p =* 0.057, HC-KOA: *p* < 0.05). In the asymmetry domain, all three groups showed similar stride length asymmetry, while both LSS and KOA showed significantly higher swing asymmetry relative to HC (HC-LSS: *p* < 0.05, HC-KOA: *p* < 0.05). Finally, in the variability domain, both LSS and KOA showed significantly higher CV of cycle duration, and swing (HC-LSS: *p* < 0.05, HC-KOA: *p* < 0.05), indicating an increase in gait variance compared to HC. LSS showed a significantly higher stride length CV than HC (*p* < 0.05), while KOA’s stride length CV was similar to HC (*p* = 0.47). Although there were no significant differences between the LSS group and the KOA group for any parameter, stride length variability demonstrated a strong trend toward a difference between the two disease groups (stride length CV; LSS 4.7 ± 0.8, KOA 4.0 ± 0.8, *p* = 0.057) with greater variability in the LSS group.

### 3.3. Minute-by-Minute Analysis of the 6MWT

Next, we performed a minute-by-minute analysis to further assess changes over time during the 6MWT. Firstly, we conducted a two-way ANOVA (three groups by 6 min) and, as expected, observed that HC had significantly different gait parameters (lower double support ratio, swing asymmetry, stride length CV, cycle duration CV, swing CV, and higher cadence, swing ratio, speed, stride length) compared to the disease groups for every minute of the 6MWT (*p* < 0.05). Only stride length asymmetry and double support CV failed to show a difference between HC and the disease groups. On a minute-by-minute basis, we observed no significant temporal change within any group.

We performed additional comparisons to evaluate the two disease groups further. Representative graphs for each domain are shown in Figure 2, and other graphs are shown in Appendix A. Focusing solely on the observed difference between the LSS and KOA groups, a significant between-group difference appeared in the middle portion of the 6MWT for stride length variability. Specifically, during the 3rd minute of the 6MWT, the stride length CV of the LSS group was significantly higher than that of the KOA group (4.29 ± 0.94 in the LSS group, 3.40 ± 0.45 in the KOA group, *p* < 0.05). In addition, the LSS group showed a trend toward higher stride length variability during the 4th minute of the 6MWT (3.94 ± 0.54 in the LSS group, 3.47 ± 0.50 in the KOA group, *p* = 0.06). This increased stride length variability in LSS indicates a worsening of gait rhythmicity.

## 4. Discussion

In this study, we derive gait parameters from foot-mounted IMU sensors worn during the 6MWT by people with LSS and KOA, compared to each other and HC for the entire 6MWT and during each minute. We characterize the findings based on four gait domains (rhythm, pace, asymmetry, and variability). When the entire 6MWT is considered, multiple gait parameters in all four domains distinguish between HC and LSS or KOA, while no statistical differences are observed between the two disease groups. Yet, the LSS group did show a trend toward higher stride length variability compared to the KOA group (*p =* 0.057). Further minute-by-minute analysis comparing gait characteristics between LSS and KOA revealed that LSS individuals displayed significantly greater stride length variability than KOA individuals during the middle portion of the walk test.

Stride length variability refers to the ratio between the standard deviation and the mean of the stride length, expressed as a percentage. Gait variability, stride-to-stride fluctuations during walking, is a potential biomarker for gait impairment and loss of gait rhythmicity [29,35,36]. An increase in variability indicates worsening gait consistency. Previous research demonstrates that maintaining gait consistency is a complex process dependent on diverse neurological structures from the cerebral cortex to the peripheral nerves and muscles [37]. Therefore, damage to any neuromuscular structure that governs locomotor control can influence gait variability [37]. Accordingly, many studies report altered gait variability (including increased CV of stride length) in various neurological disorders such as Alzheimer’s disease, amyotrophic lateral sclerosis, cerebellar ataxia, Huntington’s disease, multiple sclerosis, and Parkinson’s disease [38,39,40,41,42,43].

Similarly, LSS has a neurologic disease mechanism caused by the positional compression of the cauda equina or nerve roots, resulting in intermittent motor and/or sensory nerve dysfunction [9]. In agreement with our study findings, recent research [44] using a self-paced walking test with a chest-mounted IMU sensor also observed increased step length variability in LSS patients compared to the healthy controls. Previous research also suggested that gait irregularity can be attributed to neurogenic claudication and radicular pain, common clinical symptoms in LSS patients [44,45]. Previous research theorized that radicular pain, with its unpredictable ectopic discharge, may cause irregular walking and contribute to high gait variability [45]. Conversely, gait impairment in KOA originates from structural joint deformation that does not involve motor control mechanisms. This study’s minute-by-minute analysis for 6MWT revealed a significant difference between the LSS and KOA groups in the middle portion of the 6MWT for stride length variability. Therefore, findings from this work suggest that the stride length variability may be a potential sensitive mobility biomarker to distinguish between gait impairment from a neurological origin (LSS) and gait impairment from joint structural disease (KOA).

Findings from this work also highlighted that the middle portion of the 6MWT may be a potential window to detect subtle gait differences between individuals with different origins of gait impairment. In contrast, the first and last minutes of the 6MWT were not found to be informative for disease phenotyping. The middle portion of another long-duration gait test (400 m walking test, which is comparable with 6MWT for test duration) was recently reported to be more representative of an individual’s gait characteristics since it may provide better resolution for fatigue quantification as participants often increase gait speed during the initial and ending stages of the assessment [46,47,48]. However, previous research also demonstrated that the 6MWT could elicit a more significant fatigue effect in individuals with neurological gait impairments (multiple sclerosis) toward the end of the test [18]. Therefore, the most sensitive portion of a gait test for disease phenotyping may be disease-specific, and further investigation is needed.

There are several limitations to this study. The sample size is small, as is typical of an early investigation. More extensive studies are warranted to confirm our results. There were few minute-by-minute fluctuations during the 6MWT in any group, including healthy controls, suggesting that a six-minute walk may not provide sufficient time to detect the gait changes induced by LSS and KOA.

## 5. Conclusions

By examining gait parameters derived from foot-mounted IMU sensors worn during a 6MWT, we identified multiple parameters (in all four domains: rhythm, pace, asymmetry, and variability) that can distinguish between HC and patients with mobility limitations. Additional temporal analysis of the 6MWT on a minute-by-minute basis revealed a significantly higher stride length variability for LSS than KOA during the middle portion of the walk test. Gait variability may be a potential biomarker to distinguish gait impairment from neurological and structural joint diseases. The middle portion of the walk test may be a unique window to reveal the subtle differences observed between these conditions.

## Figures and Tables

**Figure 1 sensors-22-09301-f001:**
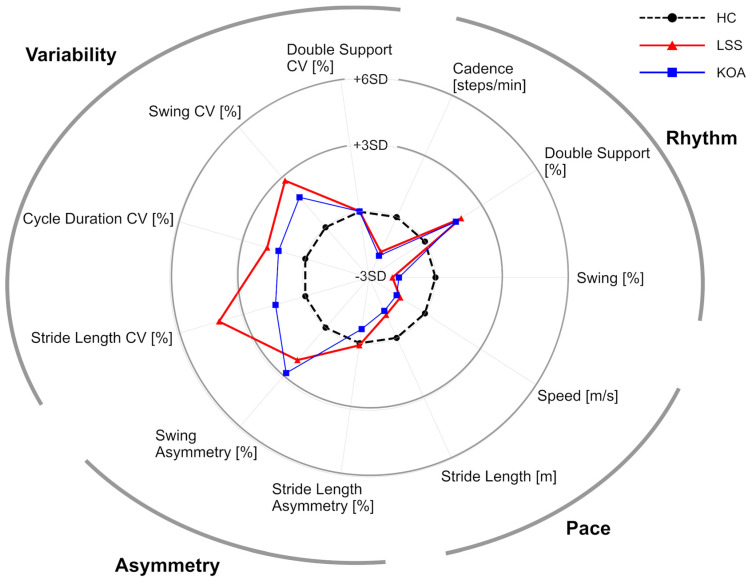
Radar plot illustrating the gait parameters extracted from the analysis of the entire 6MWT. The central black dots with connecting dashed lines represent HC data. This is compared to LSS (blue squares and lines) and KOA (red triangles and lines) with deviation along the axis radiating from the center of the plot representing the standard deviations (range; from −3 SD to +6 SD) from HC. (HC; healthy control, LSS; lumbar spinal stenosis, KOA; knee osteoarthritis, CV; coefficient of variation, SD; standard deviations).

**Figure 2 sensors-22-09301-f002:**
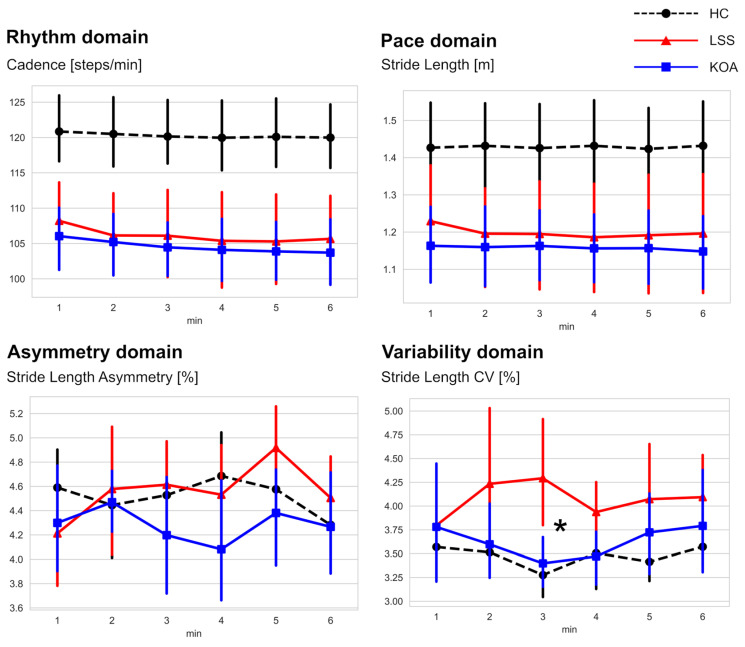
Changes in representative parameters from the minute-by-minute analysis of the 6MWT. The * identifies a significant difference between LSS and KOA (*p* < 0.05). (HC; healthy control, LSS; lumbar spinal stenosis, KOA; knee osteoarthritis, CV; coefficient of variation).

**Table 1 sensors-22-09301-t001:** Descriptions of the extracted spatiotemporal parameters.

Domain	Parameter	Unit	Description
Rhythm	Cadence	steps/min	Number of steps per minute
	Double support ratio	%	Percentage of the cycle where both feet are on the ground
	Swing Ratio	%	Percentage of the cycle during which the foot is in the air and does not touch the ground
Pace	Stride Length	meter	Distance between two successive heel strides.
	Speed	m/s	Forward stride speed of one cycle
Asymmetry	Stride length asymmetry	%	Symmetry index of stride length
	Swing asymmetry	%	Symmetry index of swing
Variability	CV for double support	%	Coefficient of variation for double support
	CV for swing	%	Coefficient of variation for swing
	CV for stride length	%	Coefficient of variation for stride length
	CV for cycle duration	%	Coefficient of variation for cycle duration

CV: coefficient of variation.

**Table 2 sensors-22-09301-t002:** Gait parameters extracted from the analysis of the entire 6MWT.

					*p* Value		
Domain	Parameter	HC	LSS	KOA	HC-LSS	HC-KOA	LSS-KOA
**Rhythm**	Cadence (steps/min)	120.3 (8.2)	106.1 (10.2)	104.6 (7.1)	**0.003**	**0.001**	0.900
	Double Support (%)	21.9 (4.0)	29.7 (5.2)	28.5 (5.9)	**0.005**	**0.019**	0.861
	Swing (%)	39.1 (2.0)	35.1 (2.6)	35.7 (2.9)	**0.005**	**0.019**	0.851
**Pace**	Speed (m/s)	1.5 (0.3)	1.1 (0.3)	1.0 (0.2)	**0.010**	**0.003**	0.878
	Stride Length (m)	1.4 (0.2)	1.2 (0.3)	1.2 (0.2)	0.057	**0.022**	0.900
**Asymmetry**	Stride Length Asymmetry (%)	4.5 (0.4)	4.6 (0.5)	4.3 (0.5)	0.900	0.416	0.340
	Swing Asymmetry (%)	4.0 (2.1)	8.1 (4.9)	9.7 (6.8)	**0.043**	**0.010**	0.797
**Variability**	Stride Length CV (%)	3.6 (0.3)	4.7 (0.8)	4.0 (0.8)	**0.003**	0.470	0.057
	Cycle Duration CV (%)	1.9 (0.7)	3.1 (0.9)	2.8 (0.8)	**0.003**	**0.025**	0.678
	Swing CV (%)	1.9 (0.5)	3.1 (1.1)	2.7 (0.7)	**0.003**	**0.025**	0.628
	Double Support CV (%)	7.1 (2.9)	7.2 (1.2)	7.2 (2.8)	0.898	0.900	0.900

Significant *p* values (*p* < 0.05) are shown in bold. (HC; healthy control, LSS; lumbar spinal stenosis, KOA; knee osteoarthritis, CV; coefficient of variation).

## Data Availability

The data of this article are available from the authors on reasonable request.

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
