# Peer review of "Gait Variability to Phenotype Common Orthopedic Gait Impairments Using Wearable Sensors"

_sensors, 2022, doi:10.3390/s22239301_

Round 1
Reviewer 1 Report
This paper presents an interesting application of wearable sensors for studying gait of patients with different impairments. The paper is well written, the analysis is strong, and the conclusions reveal interesting insights.
My only concern with the paper is that, given the topic of the special issue, there should be a little more background in the introduction that provides broader context for the specific sensors being used in the study (i.e. what other types of sensors are out there, and what are the relevant advantages / what was the rationale for selecting the sensors being used). It would also be good to have at least one conclusion that cited the relative merits of the type of sensor being utilized in the study (i.e. how did this type of sensor contribute to discovery of the insights gained).
Author Response
We thank the reviewer’s constructive feedback and revised the following section in the background section:
Traditionally, gait analysis has been conducted in well-equipped labs using 3D motion capture techniques, requiring use of costly equipment and specialized facilities. Newer technologies (including: inertial measurement unit (IMU), smartphone, low-cost video/depth camera, pressure sensor, and ambient sensors, etc.) provide alternatives that can efficiently capture and analyze movement data in various settings [20]. Among these technologies, wearable IMUs are most commonly used as the most portable and flexible option to measure mobility [21], allowing physicians, physical therapists and researchers to analyze gait in research laboratories, clinics, and in a patient’s home or community. IMU sensors contain small 3D accelerometer (linear acceleration), gyroscope (angular velocity) and/or magnetometer (magnetic field) that can be used collectively to quantify movement through various time- and frequency-domain parameters [20]. By securely attaching wearable IMU sensors to various body segments, different kinematic and biomechanical parameters can be obtained with similar validity and reliability to conventional lab-based systems [22, 23]. Gait features derived from wearable sensors have demonstrated potential clinical utility in neurological diseases by distinguishing disease populations from healthy controls and stratifying disease severity [16, 18]. However, such clinical utility has yet to be demonstrated in orthopedic research.
In conclusion, we add the following:
By examining gait parameters derived from foot-mounted IMU sensors worn during a 6MWT, we identified multiple parameters (in all four domains: rhythm, pace, asymmetry, and variability) that can distinguish between HC and patients with mobility limitations. Additional temporal analysis of the 6MWT on a minute-by-minute basis revealed a significantly higher stride length variability for LSS than KOA during the middle portion of the walk test. Gait variability may be a potential biomarker to distinguish gait impairment from neurological and structural joint diseases. The middle portion of the walk test may be a unique window to reveal the subtle differences observed between these conditions.
Reviewer 2 Report
The authors presented an analysis to identify if gait parameters representing the rhythm, pace, asymmetry, and variability domains obtained from the 6-minute walk test can phenotype gait impairment from two musculoskeletal disorders lumbar spinal stenosis (LSS) and knee osteoarthritis (OA).
- for the entire 6-minute walk:
- Significant difference between disease ground and healthy controls in gait parameters from the four domains
- no significance difference between the LSS and OA groups
- Minute-by-minute analysis:
- significance in stride length variability during the middle portion of the walk
- one of the main findings is that time series data enables precise analysis (minute-by-minute), the authors could emphasize this more in the abstract and provide background/related work in the introduction if similar works exist
- the authors could provide more descriptions of how the patio-temporal gait parameters are calculated
Author Response
We thank the reviewer’s feedback and edit the abstract/introduction as follows:
Abstract:
Mobility impairments are a common symptom of age-related degenerative diseases. Gait features can discriminate those with mobility disorders from healthy individuals, yet phenotyping specific pathologies remains challenging. This study aims to identify if gait parameters derived from two foot-mounted inertial measurement units (IMU) during the 6-minute walk test (6MWT) can phenotype mobility impairment from different pathologies (Lumbar spinal stenosis [LSS] - neurogenic diseases, and knee osteoarthritis [KOA] – structural joint disease). Bilateral foot-mounted IMU data during the 6MWT were collected from patients with LSS and KOA and matched healthy controls (N=30, 10 for each group). Eleven gait parameters representing four domains (pace, rhythm, asymmetry, variability) were derived for each minute of the 6MWT. In the entire 6MWT, gait parameters in all four domains distinguished between controls and both disease groups; however, the disease groups demonstrated no statistical differences, with a trend toward higher stride length variability in the LSS group (p=.057). Additional minute-by-minute comparisons identified stride length variability as a statistically significant marker between disease groups during the middle portion of 6WMT (3rd min: p=<.05; 4th min: p=.06). These findings demonstrate that gait variability measures are a potential biomarker to phenotype mobility impairment from different pathologies. Increased gait variability indicates loss of gait rhythmicity, a common feature in neurologic impairment of locomotor control, thus reflecting the underlying mechanism for the gait impairment in LSS. Findings from this work also identify the middle portion of the 6MWT as a potential window to detect subtle gait differences between individuals with different origins of gait impairment.
Introduction:
Numerous parameters are used to characterize gait disturbances [13-15]. These gait parameters can be classified into four domains: rhythm, pace, asymmetry, and variability [16, 17]. The rhythm domain represents measures of absolute timing. The pace domain relates to walking speed and length measures in the sagittal plane. The asymmetry domain links to differences between the right and left lower limb parameters. And the variability domain reflects fluctuations in spatiotemporal characteristics between steps. These gait parameters are often reported as mean values during the entire period of the test, however, recent work showed that the temporal change of gait performance over a period time may reveal additional subtle gait alterations that may be fatigue-induced and/or disease-specific [18, 19].
the authors could provide more descriptions of how the spatio-temporal gait parameters are calculated.
The details of the Spatio-temporal gait parameters extraction have been documented in our previous paper.
(Odonkor, C.; Kuwabara, A.; Tomkins-Lane, C.; Zhang, W.; Muaremi, A.; Leutheuser, H.; Sun, R.; Smuck, M., Gait features for discriminating between mobility-limiting musculoskeletal disorders: Lumbar spinal stenosis and knee osteoarthritis. Gait Posture 2020, 80, 96-100)
We decide to add the following in the methods section.
During the entire 6MWT, participants wore two Shimmer 3 IMU sensors (Shimmer Sensing, Dublin, Ireland) on the dorsal surface of both feet. We used previously validated algorithms to extract spatial-temporal gait parameters that reflect steady-state walking (after excluding all turning episodes) [21]. For details on gait parameter extraction, please see Supplementary file 6.
Supplementary file 6. Details on gait parameter extraction.
The Shimmer3 wearable sensor platform (Shimmer Sensing, Dublin, Ireland) was the IMU used for data collection. An IMU sensor was placed on the dorsal surface of the participant’s right and left foot using shimmer straps. Each IMU sensor consists of a 3D accelerometer, a 3D gyroscope, a 3D magnetometer. Data were sampled at 102.4 Hz and hardware synced by the control software. We used validated algorithms (a rule-based stance phase event detection algorithm) [1,2] to extract the spatiotemporal gait parameters from the IMU sensors. Prior to processing, data were resampled to 200Hz using linear interpolation to be consistent with the validated algorithms [3]. Gait cycles were detected based on the timing of two consecutive foot-flats [1]. Velocity and position of the foot were extracted by the numerical integration of the gravity-corrected acceleration data and drift corrected using the ZUPT method [4]. Heel strike and lift off angles were estimated based on the de-drifted angular velocity data [5]. Maximum angular velocity of the foot and various temporal parameters were extracted from the angular velocity signals [1]. Cycles with a turning angle between two foot-flats less than 20 degrees were considered as straight walking cycles [2].
References
[1] B. Mariani, H. Rouhani, X. Crevoisier, K. Aminian, Quantitative estimation of foot-flat and stance phase of gait using foot-worn inertial sensors, Gait Posture. 37 (2013) 229–234. doi:10.1016/j.gaitpost.2012.07.012.
[2] B. Mariani, C. Hoskovec, S. Rochat, C. Büla, J. Penders, K. Aminian, 3D gait assessment in young and elderly subjects using foot-worn inertial sensors, J. Biomech. 43 (2010) 2999–3006.
[3] W. Zhang, M. Smuck, C. Legault, M. Ith, A. Muaremi, K. Aminian, Gait symmetry assessment with a low back 3d accelerometer in post-stroke patients, Sensors. 18 (2018) 3322.
[4] E. Foxlin, Pedestrian tracking with shoe-mounted inertial sensors, IEEE Comput. Graph. Appl. (2005) 38–46.
[5] B. Mariani, S. Rochat, C.J. Büla, K. Aminian, Heel and toe clearance estimation for gait analysis using wireless inertial sensors, IEEE Trans. Biomed. Eng. 59 (2012) 3162–3168.mbs).